# Prepartum Vaccination Against Neonatal Calf Diarrhea and Its Effect on Mammary Health and Milk Yield of Dairy Cows: A Retrospective Study Addressing Non-Specific Effects of Vaccination

**DOI:** 10.3390/ani15020203

**Published:** 2025-01-14

**Authors:** Caroline Kuhn, Holm Zerbe, Hans-Joachim Schuberth, Anke Römer, Debby Kraatz-van Egmond, Claudia Wesenauer, Martina Resch, Alexander Stoll, Yury Zablotski

**Affiliations:** 1Clinic for Ruminants with Ambulatory and Herd Health Services, Centre for Clinical Veterinary Medicine, Ludwig-Maximilians-University, 80539 Munich, Germany; h.zerbe@lmu.de (H.Z.);; 2Institute for Immunology, University of Veterinary Medicine, 30559 Hannover, Germany; hans-joachim.schuberth@tiho-hannover.de; 3Mecklenburg-Vorpommern Research Centre for Agriculture and Fisheries, Institute of Livestock Farming, 18196 Dummerstorf, Germany; a.roemer@lfa.mvnet.de; 4RinderAllianz GmbH, 17348 Woldegk, Germany; dkraatz@rinderallianz.de (D.K.-v.E.);; 5Intervet Deutschland GmbH, 85716 Unterschleissheim, Germanyalexander.stoll@msd.de (A.S.)

**Keywords:** transition cow, vaccination, trained immunity, cattle, bovine mastitis, epidemiology

## Abstract

Prepartum vaccinations of cows are performed to boost colostral antibodies available to calves in the first weeks of life. While beneficial effects of the vaccine for calves are well known, less has been investigated on its effects on the cow during the sensitive period around calving. There is evidence that vaccines have non-specific effects, including altered susceptibility to infectious diseases other than the targeted pathogens and involve the innate immune response of the vaccinated animal. Therefore, this study analyzed data from 73,378 cows on 20 farms in Germany to evaluate the effects of prepartum vaccination against neonatal calf diarrhea on mastitis, somatic cell count, and milk production. The results demonstrated that herd management had the highest influence, while prepartum vaccination had no significant impact on mammary health and milk yield. These findings suggest that prepartum vaccination does not compromise mammary health or milk productivity. Further research is recommended to explore non-specific vaccine effects on other organ systems, diseases, and production metrics in dairy cows.

## 1. Introduction

Prepartum vaccinations of cows are primarily performed to protect calves from infectious diseases, such as neonatal calf diarrhea (NCD), in their first weeks of life. For this purpose, the cow is vaccinated during the last weeks of pregnancy to induce an adaptive immune response with high levels of pathogen-specific antibodies and obtain enriched colostrum for additional protection of the newborn calf [1,2,3]. Prepartum vaccination of cows is regarded as safe and usually no adverse reactions are reported [4,5]. Adverse reactions are known as any harmful or unintended outcomes that are directly caused by the vaccine or its components, including hypersensitivity, fever, local swelling, and short-term reactions such as increased levels of acute phase proteins. These reactions can differ between live and non-live vaccines, as well as between adjuvants administered [6,7,8]. In contrast to the adverse effects of vaccination, which are harmful, there are vaccination-induced effects beyond antigen-specific induction of adaptive immune responses, known as non-specific effects (NSEs) [9]. These can be beneficial or harmful for the individual. However, underlying mechanisms are intensely discussed [10], and possible characteristics of live, attenuated, or killed vaccines in this regard are not fully elaborated [9]. In cattle, it has been demonstrated that the immediate innate response to killed vaccines resulted in an altered gene expression of circulating immune cells [11,12]. While live vaccinations in humans are central to ongoing investigations, there is a lack of research on the NSEs of veterinary vaccine products [9]. Progress has been made in understanding NSEs, particularly in rodents and humans [13]. The inflammatory response after active vaccination can lead to an altered functionality of circulating immune cells. The innate response to vaccination during pregnancy has been demonstrated in mice where the development of the fetal immune system was modified in utero by epigenetic changes [14]. This suggests that vaccination-induced epigenetic modification also alters the immune reactivity of the vaccinated mother. Such vaccination-induced, innate-response-mediated effects can lead to a status called innate immune memory, based on epigenetic modifications [15]. Such effects have been well-studied for certain vaccines, such as the Bacille Calmette Guerin (BCG) vaccine, which contains an avirulent Mycobacterium bovis strain [16,17]. This vaccine resulted in a trained functional phenotype (trained immunity) of circulating myeloid cells in calves, referring to primary stimulus-augmented innate immune responses towards a secondary stimulation [15].

Whether contemporarily used veterinary vaccines, especially those applied to cows in the transition period during late pregnancy, have non-specific effects or not, has not been addressed so far. A number of studies explored variable effects of cow vaccinations with conventional vaccines on milk yield. No effect on milk yield was reported after vaccination with a core antigen vaccine against Gram-negative bacteria [18]. Lower milk yields were shown after vaccination with an inactivated Coxiella burnetii vaccine [19], whereas heifers vaccinated with a BCG vaccine showed higher milk yields [20]. How these effects were mediated was not addressed so far and, to the best of our knowledge, no study addressed the potential NSEs of prepartum vaccination against NCD on the prevalence of postpartum mastitis and the milk yield of cows. Therefore, the aim of this retrospective cross-sectional study was to analyze the effect of contemporary prepartum vaccines on mammary health and milk yield of the periparturient cow. It is hypothesized that the prepartum vaccination against NCD induces beneficial non-specific effects, which allow the dysregulated immune function of the periparturient cow to better cope with mammary infections. These effects would be measurable through improved mammary health outcomes and production metrics during the very sensitive peripartum period.

## 2. Materials and Methods

### 2.1. Study Population and Farm Selection

Data were obtained from test herds of the RinderAllianz (RA), a breeding organization that supervises numerous dairy farms in Mecklenburg-Vorpommern, Saxony-Anhalt, and Brandenburg. The service includes sperm sales, mating, insemination, cattle marketing, milk control, and analyses. In cooperation with the RA, comprehensive data from 20 test farms were obtained in the context of this work. At the timepoint of data collection in June 2021, prepartum vaccination was performed on ten of the participating farms. Milk yield was balanced, so that each group of farms contained five high yielding farms with mean energy corrected milk yield in 305 days of lactation (ECM 305) above 11,000 kg and five low yielding farms below 11,000 kg ECM 305. Informed consent was obtained from all participating farms. All of these farms share a common history as former Agricultural Production Cooperatives in former East Germany until 1989. Altogether, data of these 20 farms comprises herd records, reproduction data, milk recordings, health documentation, and holding registers of 73,378 dairy cows of 22 herds located across all three operating regions of the RA between January 2007 and September 2020.

### 2.2. On-Farm Data Collection

Each farm was visited on-site to collect contextual information and conduct an in-depth survey. The survey comprised information on vaccination history, dry-off management, housing systems, health management, monitoring during calving, milking and colostrum management, hygiene and feeding management. On-site visits took place between October 2021 and August 2022 with the herd managers. Depending on the size, infrastructure of the farms, and distribution of tasks between workers, responsible persons were consulted, if necessary. The questionnaire was always filled out by the same surveyor in order to ensure conformity of documentation. One farm was surveyed online due to pandemic precaution-measures while all others were visited on-site, allowing a deeper understanding of the conditions of the location. Participating farmers were very cooperative and, with no exception, showed the surveyors around all relevant areas of the facility, the housing of dairy cows in all stages of the production cycle—calves, heifers, dry cows, fresh and late-lactating cows-, milking installation, and the calving area, which was especially inspected carefully. Apart from data collection, the purpose of the visits ensured that at least no obvious problems in management and hygiene are apparent on the respective farms. Furthermore, the local regularly visiting veterinarians were consulted to better estimate herd management. Depending on time availability, some farm visits were attended in-person by the veterinarians, in other cases, they were consulted by phone, and in all cases, the reliability of health management was confirmed.

### 2.3. Data Pre-Processing

Step-by-step data were joined and cleaned according to the needs of the study using the software R version 4.3.1 and R Studio (Figure 1). Herd record data were inspected for inconsistencies before being converted into a format where each observation corresponds to one lactation, whereby chronological adaptions were made. Events during the dry period are assigned to the subsequent lactation, in order to tailor the database to the transition period of the cow. Reproduction data of 148,268 lactations, milk yields, and components from 1,561,273 recordings and 1,298,703 diagnoses from health documentation were added. Moreover, on-site survey data were manually transferred from paper-based forms filled out on-site into ExcelTM-sheets and further integrated into the final database. Observations were excluded when on-site data or health documentation were not available. Information on vaccination management on each farm was assigned to the vaccination status on an individual cow basis and transition period. All observations with other vaccinations other than prepartum vaccination against neonatal calf diarrhea were excluded to reduce interaction with other vaccines. When a farm changed its vaccination management or did not vaccinate for a distinct time period, a buffer of one month (15 days prior to and 15 days after the date of change) was created, thus risk of false entries is minimized. Here, farm-specific time of vaccination was paid attention to in order to correctly assign vaccination periods and calving dates.

Originally 73,378 dairy cows in 22 herds across 20 farms were provided for this study, of which 53,370 dairy cows in 21 herds across 19 farms could be further included for analysis after matching herd records with health documentation, reproduction data, milk records, and the on-site survey. This first dataset consisted of 120,394 transition periods in total. The second and smaller dataset 2 represents a subset of dataset 1, containing 1002 transition periods from 1002 primiparous, utmost healthy cows from four alternately vaccinated herds. Dataset 2 allows analyses on rather ideal conditions while presuming that health and immunological status of primiparous cows differ from multiparous cows [21,22]. Furthermore, it was intended to reduce the influences of pathologic processes by excluding diseased cows. Regarding mammary health, cows diagnosed with mastitis were excluded, taking into account that mastitis can lead to lower milk yields [23]. As a somatic cell count (SCC) below 100,000 cells is considered physiological [24,25], lactations were only included when SCC of the first test day of milk recordings were below this threshold. Risk of ketosis and diagnosis of retained placenta and metritis were also eliminated from dataset 2. In alternately vaccinated herds, a timeframe of twelve months before and twelve months after change in vaccination management (from vaccination to non-vaccination or vice versa) was isolated. By this means, the comparison of vaccinated and non-vaccinated transition periods was less influenced by calving year.

### 2.4. Description of Vaccination Related Parameters

A supplementary table provides an overview of response and independent variables, definitions, composition, and values (Appendix A). Three vaccines were used:Bovilis^®^ Rotavec^®^ Corona (Intervet Deutschland GmbH, Unterschleissheim, Germany) (RC): *n =* 27,769, containing inactivated bovine rotavirus (serotype G6 P5), inactivated bovine coronavirus (strain Mebus), *E. coli* (K99 Antigen), mineral oil, and aluminum hydroxide.Scourguard^®^ 3 (Zoetis Deutschland GmbH, Berlin, Germany) (SG): *n =* 8352, containing live attenuated bovine rotavirus (strain Lincoln), live attenuated bovine coronavirus (strain Hansen), inactivated *E. coli* (K99 Antigen), Alhydrogel.Bovigen^®^ Scour (Forte Healthcare Ltd, Dublin, Ireland) (BS): *n =* 8004, containing inactivated bovine rotavirus (serotype G6 P1), inactivated bovine coronavirus (strain C-197), *E. coli* (K99 Antigen), Montanide ISA 206 VG.

RC and BS are applicated once and SG is applicated twice, whereby the date of the first application was considered the time of vaccination. In 18,453 cases, the vaccine product could not be associated (not specified). On the basis of the used vaccine products, the variable adjuvant was designed grouping animals in those vaccinated with alum (RC, SG) or montanide-containing (BS) vaccines.

### 2.5. Description of Mammary Health and Milk Yield Parameters

Each two response variables were elaborated to represent milk performance and mammary health: *ECM 305* defines the energy corrected milk yield in 305 days of lactation and *ECM FTD* on the first test day of milk recording. For this, the milk recording parameters were consulted to calculate as follows [26]:(1)$$ECM=milk\;yield\;\left(kg\right)\times \frac{0.38\times fat\left(\%\right)+0.21\times protein\left(\%\right)+1.05}{3.28}$$

To approach an adequate representation of the mammary health status of the cows, parameters of different origins were consulted. Farm health documentation was transformed into the binary variable mastitis indicating the prevalence of mastitis diagnoses of each cow when the diagnosis mastitis was registered at least once within the time period of 10 days p.p. The analysis was completed with SCC as a further stringent variable, less influenced by personnel or documentation. It is defined as the somatic cell count on the first day of milk recording and was logarithmically transformed to stabilize variance. SCC is seen as a good parameter to identify intramammary infections [27] and is considered physiological below 100,000 [24,25]. The farms conducted milk recordings once a month. Thus, the first test day of milk recordings and therefore Days in Milk (DIM) at the time of data collection of the variables ECM FTD and SCC differs across the observations, resulting in limitations for these variables as they represent only one single point of time during the first weeks of lactation where the milk yield and cell count vary. The mean DIM was considered between the groups investigated, ascertaining comparability.

Eligible influencing variables were defined, firstly the above-mentioned vaccination-related variables, and further, farm management and cow-related variables. Management-related variables encompass the following variables: The herd itself representing the general farm management. Calving year denotes the year when cows calved, potentially impacting health due to environmental or management changes. The variables herd and calving year might not be understood in isolation but rather in combination. Although e.g., climate or pathogen diversity might vary between the years, the fluctuations of the years are probably more subject to farm management than the year itself. For this reason, they were consequently combined as random effects in the statistical models. Herd Size indicates the annual total number of cows in the herd, potentially influencing disease spread and herd dynamics. The variable was transformed from numeric to character type by segmenting the categories into small, medium and large sections with the binning-function of the dlookr-package. The quantile method was employed to determine break points, ensuring an equitable distribution of farm sizes across the derived categories. Herd replacement rate of the previous year, affecting overall herd health and productivity in the respective year, was calculated as percentage of the number of primiparous calvings, compared to the number of secondi- or multiparous calvings. Access to pasture reflects the availability and quality of grazing areas during the dry period, impacting especially the exercise level of the cow during calving and lactation. Flooring refers to the type of flooring within the barn during the transition period, impacting cow comfort and hoof health with the two categories deeplitter or slatted floor. Score of hygiene indicates the cleanliness level of the cows and density of possible pathogens. A score between 1 and 4 was documented on the date of on-site survey [28]. Calving box describes the two options of calving in a group or in an individual box, influencing behavioral dynamics around calving. When calving in individual boxes, pen change was conducted during transition period, which can affect the cows’ metabolism in combination with prepartum vaccination [29]. *Milking Frequency* defines how often cows are milked daily, affecting mammary health. Here, robot-milked cows could not be traced for this variable. Moreover, Type of dry-off describes the method used when cows are dried off from milking, impacting mammary gland health by using or not using antibiotic dry-cow therapy. Additionally, supplements such as *Energy Supplements*, *Calcium Supplements*, *Vitamin D3 Supplements*, and further *Vitamin and Trace Element Supplements* and *Monensin* given shortly before, during, or shortly after calving were provided, possibly impacting nutritional and immune status. The following aspects were assigned to cow related variables: Cows were categorized as primi-, secondi-, or multiparous in the variable parity. Length of the dry period was provided partly in weeks during the on-site survey and calculated in days and can therefore deviate slightly. The first lactation age was calculated as the difference between calving date of first calving and the cows’ birthdate, providing the following categories: low first lactation age therefore is less than or equal to 700 days, medium between 701 and 750 days, and high more than 750 days, taking into consideration that milk yield and ingredients can differ between these groups [30]. The four calving seasons enable us to investigate potential seasonal variations in dairy cow health, while spring is defined as calving date between March and May, summer between June and August, autumn between September and November, and winter between December and February. This variable not only refers to the calving itself, but also allows us to explore whether specific health events were more prevalent during certain times of the year. Finally, risk of ketosis provides a conclusion on the ketotic metabolic status, determined by the milk components on the first day of milk recording. Ketotic risk was assumed if the fat-protein-ratio exceeds 1.4 and the lower limits of protein content are undercut or upper limits of fat content are passed. Limits were calculated according to Glatz-Hoppe et al. [31]. Assessing the risk of ketosis in this critical period was aimed at predicting and mitigating potential metabolic disruptions affecting the cows’ health.

### 2.6. Statistical Analyses

The software R version 4.3.1 and R Studio were used for statistical analyses [32]. Descriptive analysis of dataset 1 for the categories transition periods without vaccination during the dry period (NON VACC) and transition periods with prior vaccination (VACC) was undertaken. All eligible variables were examined for any possible influence on the response variables mastitis, *SCC*, *ECM FTD*, and *ECM 305* in univariable analysis. The variables in question were further checked for missing value patterns. Generalized mixed-effects models were executed using the lme4- package. The variables herd and calving year were considered as nested random effects. Given the nested structure of the data involving herds and calving years, different combinations of random effects in mixed-effects models were assessed to determine the most suitable formulation based on Akaike’s Information Criterion. The *p*-value threshold was adjusted in accordance with Goods’ recommendations for large number of observations [33]. By applying the function$$f\left(x\right)=\frac{0.05}{\sqrt{\frac{x}{100}}},$$
the *p*-value of each model outcome could be evaluated, resulting in significance thresholds between 0.0158 for dataset 1 and 0.0014 for dataset 2. All variables that were significant in the univariable generalized mixed-effects regression of the response variables mastitis and *SCC*, according to Goods’ *p*-value threshold, were further included in the subsequent multivariable analysis. When preparing the multivariable analysis for the response variables ECM FTD and ECM 305, it soon became clear that variables other than the vaccination-related parameters overlapped with the results. Furthermore, missing value patterns foreclosed multivariable analysis. Therefore, dataset 2, a subset of dataset 1, was elaborated, reducing influencing variables. The available disease-associated variables, mastitis, *SCC* above 100,000, retained placenta, metritis, and ketotic risk were used to exclude diseased cows, as these are known influencing factors on milk yield [34,35]. Parity, another known influencing variable on milk yield and the periparturient cow’s immune system [22,36,37] proved to be significant in the univariable analysis of the present study. Thus, parity was reduced to primiparous cows. Furthermore, the pairing of VACC and NON VACC on four alternately vaccinating farms was conducted to further control confounding effects. In addition, only data in the period of twelve months before and after change in vaccination management was selected to minimize the influence of calving year. With dataset 2, containing primiparous, utmost healthy cows from alternately vaccinated herds, univariable and multivariable analysis was carried out. Quantile regression was conducted to model the relationship between predictor variables and median of the response variable. Additional broadening of the analysis with the random forest-algorithm [38] was conducted in order to obtain a ranking of importance of influencing variables for mastitis and *SCC*, including herd and calving year, previously inserted as random effects.

## 3. Results

### 3.1. Descriptive Summary: Vaccination Management in Participating Herds and Key Figures

Dataset 1, consisting of 120,394 transition periods could be divided into 57,166 NON VACC and 63,228 VACC. These transition periods were distributed across non-vaccinated, continuously vaccinated, and alternately vaccinated herd (Table 1).

Despite retrospectively adapting the groups according to the specification of vaccination periods through on-farm data collection, NON VACC and VACC are comparable in mammary health and milk yield parameters (Table 2). The overall mastitis prevalence was 6.2% (VACC 6.6%, NON VACC 5.8%). Overall median SCC was 69,000 (VACC 73,000; NON VACC 65,000). In VACC cows, the median ECM FTD was 37 kg and ECM 305 was 9739 kg, compared to 36 kg ECM FTD and 9597 kg ECM 305 in NON VACC cows. The overall median ECM FTD was 37 kg, ECM 305 was 9674 kg. Dataset 2 represents a subset of dataset 1, containing 1002 transition periods from 1002 primiparous, utmost healthy cows in four alternately vaccinated herds, thereof 525 NON VACC and 477 VACC. In dataset 2, the overall median SCC is 49,000 (VACC 50,000; NON VACC 47,000), ECM 305 8748 kg (VACC 8797 kg; NON VACC 8696 kg), and ECM FTD 29.9 kg (VACC 30; NON VACC 29.8).

### 3.2. Prepartum Vaccination Was Not Significantly Associated with Mammary Health Parameters

To identify significant influencing parameters for the response variables mastitis and SCC, univariable mixed-effects logistic regression was performed, taking into account the available meaningful variables. In both analyses, the likewise significant variables, herd and calving year, were applied as random effects in a nested structure. Mastitis was significantly associated with parity, calving season, and access to pasture. For the SCC, the variables parity, calving season, flooring, and farm size emerged as significant. All of these significant variables were further included for multivariable mixed-effects regression to investigate the influence of prepartum vaccination on mastitis and SCC. Multivariable analyses revealed that prepartum vaccination had no significant influence on both mastitis prevalence (Table 3) and SCC (Table 4).

### 3.3. Prepartum Vaccination Was Not Significantly with Milk Yield in Healthy Primiparous Cows

Univariable linear mixed-effects models showed higher ECM FTD and ECM 305 in VACC, compared to NON VACC (Appendix A). Multivariable analysis was not possible with the same dataset due to missing values resulting from the initial joint of herd and milk records. As a solution, dataset 2 was created to reduce the number of missing values and control confounding variables. Here, available disease-associated variables, such as *mastitis*, *SCC* above 100,000, *retained placenta*, *metritis*, and *ketotic risk* were used to exclude diseased cows. Further, only four herds that were vaccinated alternately were taken into account to allow for a comparison between VACC and NON VACC cows in more identical environments. Additionally, the data were limited to the time period of twelve months before and after change in vaccination management. With primiparous, utmost healthy cows from alternately vaccinated herds, univariable and multivariable analysis was performed. No significant influence of prepartum vaccination on milk yield parameters could be confirmed in these more ideal conditions, neither in uni- nor in multivariable models. However, the variable *replacement rate* remained significant for ECM 305 and ECM FTD, as well as the rest period for ECM 305 (Table 5).

Quantile regression models for four individual herds clarified that other factors, such as *herd* itself, have a higher influence on milk yield parameters than the prepartum vaccination status in healthy primiparous cows (Figure 2).

### 3.4. Herd Management Related Factors Were Most Relevant for Mammary Health

In order to broaden the analysis for the mammary health parameters *mastitis* and *SCC*, random forest-analysis was performed. With this machine-learning algorithm, influencing variables were ranked by importance, allowing for the comparison between all variables, including *herd* and *calving year*, which were applied as random effects in previous analyses of the study. Findings suggest that herd management-related parameters are the most relevant influencing factors for both response variables, either directly as *herd* or indirectly as *calving year* and *farmsize*. *Parity* proved to be among the top three influencing variables. Prepartum vaccination status, however, takes the last or second last place in this ranking (Figure 3).

## 4. Discussion

The high incidence of postpartum infectious diseases of the cow poses challenges for the current dairy industry. During the transition period, defined as three weeks before until three weeks after parturition [39,40], the dairy cow’s health is challenged and typical disease symptoms accumulate. Mastitis, metritis, ketosis, digestive disorders, and laminitis have their highest incidences during early lactation [41,42,43]. The mammary gland is particularly susceptible to pathogens during colostrogenesis [44]. Mammary disorders negatively affect animal welfare, milk yield, and the financial situation of the dairy farm, especially in early lactation [23]. Reasons for mammary disorders may originate from a poorly regulated, dysregulated, or suppressed immune system [11,45,46]. Therefore, ways are needed to modulate the immune system of the transition cow to cope better with infectious pathogens. An attractive possibility could be a vaccine-induced modulation of the immune system. The vaccination-induced mediator release after an initial activation of innate immune processes may have mid- or long-term effects on subsequent immune mechanisms depending on the duration of mediator-mediated epigenetic modifications in various cell types [15]. Although vaccine-mediated epigenetic alterations were reported for distinct vaccines, it remains unknown whether contemporarily used prepartum vaccinations against NCD are able to induce such NSEs in cows.

In this context, it was investigated whether the prepartum vaccination against NCD of pregnant cows has an impact on the prevalence of mastitis, the somatic cell count, and the short- and long-term milk production post-partum (p.p.). In univariable analysis, no significant associations between prepartum vaccination and mammary health could be found, but higher ECM FTD and ECM 305 in VACC was observed. However, multivariable analysis clearly showed that prepartum vaccination had no effect, neither on the mastitis prevalence (Table 3), the SCC (Table 4), nor the milk yield (Table 5). These findings mirror, in part, those of Scott et al. [18], who found no effect on milk yield after vaccination with a core antigen vaccine against Gram-negative bacteria, although it contained a dedicated immune-enhancing/-modulating adjuvant (ENDOVAC-Dairy^®^ with Immune Plus, Endovac Animal Health, Columbia, USA). Our findings argue against a non-specific effect of prepartum cow vaccination with the NCD vaccines applicated. Such non-specific effects were reported by others after vaccination of heifers with a live-attenuated BCG-strain [20]. Vaccination with this well-characterized, non-specific effect-inducing live vaccine resulted in higher milk yields in the first 100 days p.p. Another study reported on a farm-specific reduction in mastitis incidence after prepartum intranasal vaccination of dairy cows with a virus live vaccine [47]. The use of inactivated killed vaccine in dataset 2 suggests that the attenuated strains in the NCD vaccines may not have been able to induce the same mechanisms as the BCG vaccine or the intranasally administered modified live vaccine against BRSV and PI3. The influence of vaccine type and adjuvant was evaluated only in udder health analysis, where two different killed vaccines and one live vaccine with different adjuvant combinations were applied. However, no significant effects were observed. These findings are in contrast to Cortese et al. [47], where milk yield losses were reported after vaccination with an inactivated Coxiella burnetii vaccine containing no dedicated adjuvant [19]. Thus, the interaction of the cows with Coxiella-derived molecules could have induced a response leading to an altered secretion or synthesis capacity of mammary epithelial cells and strengthen the hypothesis that vaccine-induced mechanisms can have non-specific effects in cows. The comparison of the three mentioned studies with ours is limited as methodologies differed. In addition to different vaccine types and used adjuvants, prepartum vaccination was only performed by Retamal et al. [20], whereas Scott et al. [18] and Schulze et al. [19] vaccinated cows during lactation. A closer look at statistical methods and thorough examination of multiple predictors is substantial to reveal possible confounding events. In the present study, methodological reduction to univariable analysis would have resulted in higher milk yields of vaccinated cows, leading to spurious correlation.

Through multivariable analysis, the effect of prepartum vaccination on milk yield was more accurately determined by excluding disease-associated variables and limiting the analysis to primiparity, which are commonly addressed variables in the analysis of herd records from transition cows [22]. Pairing lactations by matching VACC and NON VACC from the same herd allows for a more confident attribution of the differences between the groups to the actual exposure. Quantile regression provided insight into the interactions with other predictors. Here, the difference in milk yield between the herds was higher than between VACC and NON VACC. Similarly, the effect of prepartum vaccination had opposite effects depending on the herd. The random forest machine learning-algorithms, a model robust to multicollinearity of variables, facilitated the ranking of importance of influencing factors. While *herd* and *calving year* were considered random effects in regression models, random forest-analysis makes them comparable with other influencing variables. This study concludes that there were no significant effects of prepartum vaccination on milk yield. In the study of Retamal et al. [20]—according to the author’s assessment—these other influencing factors cannot be ruled out, as rather simple analysis was performed on milk yield variables with contingency tables and the Wilcoxon–Mann–Whitney test. In Schulze et al. [19] and Scott et al. [18], similar statistical methodologies as in the present study were applied with linear mixed-effects models. Multiple influencing variables were considered; therefore, results are statistically more comparable to Schulze et al. [19] and Scott et al. [18] but still limited, according to the different study designs. Generalized linear mixed models have proven to be a good and flexible method, especially for transition cow analysis [22].

A large number of observations lay the foundation for statistical power, enhances the precision of analysis, and enables thorough investigation of subgroups and variables to control confounders. Simultaneously, large datasets entail several challenges. While even small effects can be discovered, the probability of type I errors and therefore false positive significant effects is increased. For this reason, the *p*-value threshold was adjusted in accordance with Goods’ [33] recommendations, resulting in higher hurdles of significance. Herd records and their retrospective nature, especially diagnostic data, bear the potential for documentation variability. On all farms, health management is conducted in close collaboration with the attendant veterinarians, diagnostics of acute diseased animals are performed by veterinarians, but identification of frequently occurring diseases, such as mastitis, is usually subject to standard operating procedures of the farm and subsequently documented by managing or milking staff. Therefore, misclassification and differences in documentation between the farms, and, respectively, herds, cannot be ruled out and might compromise accuracy and consistency. Hence, there is a risk that, for instance, high numbers in mastitis documentation on a farm do not inevitably represent actual high occurrence of mastitis but originate in precise documentation of e.g., subclinical mastitis. Precise documentation might lead to false high diagnostic frequency, whereas not recognized diseases due to farm management or inconsistent health documentation might lead to false low incidences. The discussion also considered whether the overall quality of farm management influenced the decision to vaccinate cows. It is unclear which direction this potential bias may take. On one hand, farms with excellent health management and superior animal health status may choose to vaccinate cows as a precautionary measure. On the other hand, farms with poorer health management may vaccinate cows to address the animals’ poor health status.

Recognizing these possible variances across herds and risks of biased data as limitations of this study, comprehensive data pre-processing and validation procedures were implemented. First, to gain a deeper understanding of the data and verify the comparability of herds, on-site farm visits were conducted. These visits were usually accompanied by the local veterinarians who could provide an assessment of herd management over the past years and therefore beyond the time of the onsite-survey. It is noteworthy that all included farms share a common history as former Agricultural Production Cooperatives in the German Democratic Republic until 1989, making them more comparable than other farms in Germany but not entirely eliminating heterogeneity in management practices. Secondly, an exploratory analysis of herd management-related variables was conducted for each herd using contingency tables to better differentiate between systematic and random errors. Thirdly, the diagnostic documentation for each herd across years and months was examined, comparing different diagnostic results with overall patterns to understand and interpret the trends in health documentation and anticipate documentation variability. However, the retrospective nature of the data and the reliance on farm-level documentation practices remain limitations that could affect accuracy. On some of the farms, more detailed diagnostic data are available with subordinate terms, such as the subdivision of mastitis into forms of inflammation or pathogen etiology. It was agreed upon the utilization of the generic term mastitis as the least common denominator of diagnostic health documentation across all herds. After excluding inconsistent time periods (e.g., initial phase/first year of health documentation in a herd, allowing time for familiarization), diagnostic health documentation of mastitis was assessed reliable for analysis. Findings of overall exploratory analysis suggest that the variables *herd* and *calving year* inherit high influence on *mastitis* and other variables, consequently these were included as random effects in the subsequent linear mixed-effects regression. Nevertheless, the potential for unmeasured confounders cannot be completely ruled out. Additional to mastitis, further robust and objective variables, which are less susceptible to variations in documentation, were incorporated. Originating from automated milk testing, *SCC* and milk yield parameters are hardly affected by humans. Thus, the authors regard such variables as more robust than *mastitis*. Given that milk recordings occur at monthly intervals on the respective farms, the variability in DIM among data points raises concerns about the comparability of the groups. To address this issue, the average milk performance was investigated in relation to the average DIM on the day of milk testing. Our findings suggest that the observed increase in average milk performance does not exceed the contrast estimates obtained through univariable analysis. As a result, it was concluded that the differing timing of data collection did not significantly impact the results. Despite these measures, the retrospective nature and inherent variability of the dataset introduce some limitations to the study’s explanatory power. Moreover, the scope of the study was limited to three specific vaccine products and to the mammary gland, which may not capture all effects of other prepartum vaccinations or impacts on other organ systems.

The findings underline the importance of herd management-related factors. This work provides insights from a large database in the field, particularly for the German dairy industry. The focus is on study design and statistical methodology. Further research is needed to explore the potential impact of vaccination on other infectious diseases, as well as any correlations with other organ systems and production metrics

## 5. Conclusions

The prepartum vaccination of cows with neonatal calf diarrhea using three different commercial products did not affect milk production, somatic cell count, and the incidence of mastitis. Thus, this study does not provide significant evidence for NSE of prepartum maternal vaccination against NCD for cow mammary health and milk yield parameters. It remains to be further elaborated whether vaccination ingredients, such as live or attenuated vaccines, as well as adjuvants, play a role in NSEs.

## Figures and Tables

**Figure 1 animals-15-00203-f001:**
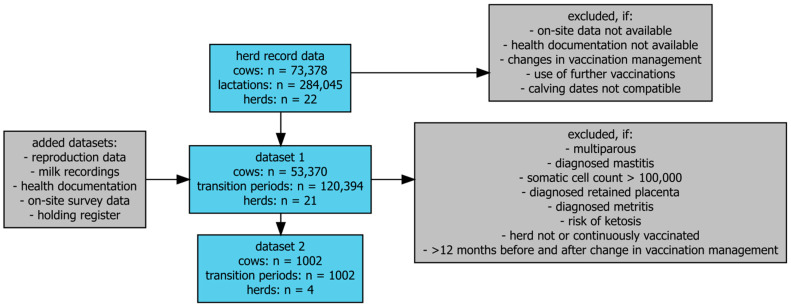
Simplified flow diagram of the studies’ data pre-processing.

**Figure 2 animals-15-00203-f002:**
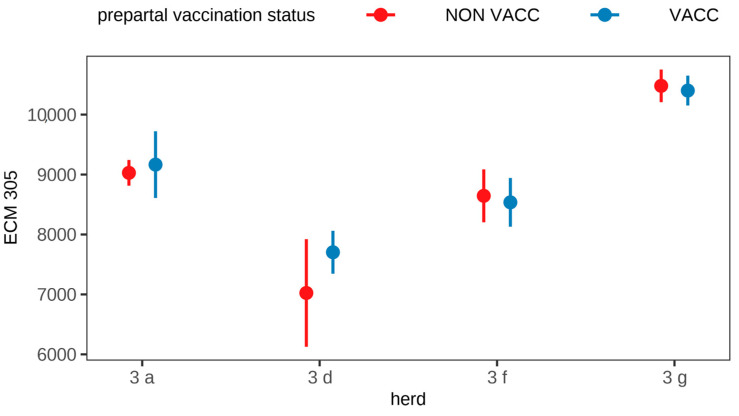
Median energy corrected milk yield in 305 days of lactation (ECM 305) of four herds (Table 1: 3a, 3d, 3f, 3g), comparing prepartum vaccinated (VACC) and non-vaccinated (NON VACC) cows in quantile regression. The error bars display 95% confidence intervals.

**Figure 3 animals-15-00203-f003:**
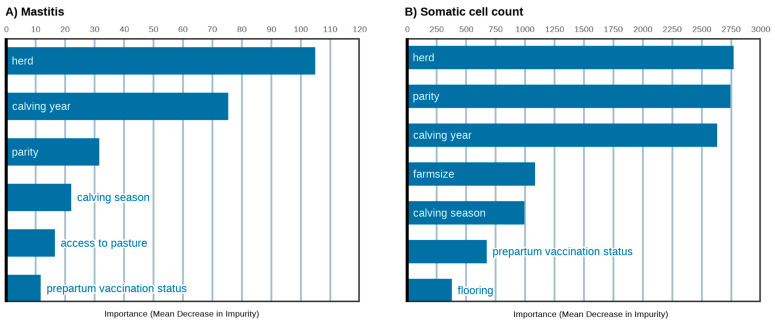
Ranking of importance of influencing variables on (**A**) mastitis prevalence and (**B**) somatic cell count by a random forest model. The importance of the predicted variable is represented by the mean decrease in impurity, which is the reduction in uncertainty in the model. Importance is associated with the ability to predict the response variable.

**Table 1 animals-15-00203-t001:** Numbers of transition periods in individual herds with different vaccination management.

Dataset 1		
	**NON VACC ^a^** **(*n =* 57,166)**	**VACC** **^b^** **(*n =* 63,228)**
Non-vaccinated herds ^c^		
1a	8791	0
1b	4459	0
1c	1902	0
1d	3531	0
1e	821	0
1f	636	0
1g	3231	0
1h	5476	0
Continuously vaccinated herds ^d^		
2a	0	1305
2b	0	2387
2c	0	8093
2d	0	7429
2e	0	18,453
Alternately vaccinated herds ^e^		
3a	1744	1545
3b	3128	5807
3c	4137	227
3d	969	6885
3e	1478	327
3f	9182	5555
3g	5847	4007
3h	1834	1208
**Dataset 2**		
	NON VACC ^a^ (*n =* 525)	VACC ^b^ (*n =* 477)
Alternately vaccinated herds ^e^		
3a	159	109
3d	137	142
3f	141	141
3g	88	85

^a^ no vaccination during the dry period; ^b^ vaccination between 2.5 and 8 weeks before expected calving date; ^c^ herds without vaccinations within the dry period; ^d^ herds with continuous prepartum vaccinations; ^e^ herds with prepartum vaccinations of parts of the herd or during defined time periods of time.

**Table 2 animals-15-00203-t002:** Descriptive summary—mammary health and milk yield in herds with different prepartum vaccination statuses.

Dataset 1			
		**Prepartum Vaccination Status**	
	**Overall** **(*n =* 120,394)**	**NON VACC ^a^** **(*n =* 57,166)**	**VACC ^b^** **(*n =* 63,228)**
Mastitis, %	6.2%	5.8%	6.6%
SCC ^c^, median (IQR ^d^)	69 (34–184)	65 (32–174)	73 (36–196)
Unknown	27,839	10,136	17,703
ECM 305 ^e^, median (IQR ^d^)	9674 (7884–11,236)	9597 (7851–11,352)	9739 (7914–11,161)
Unknown	5431	2648	2783
ECM FTD ^f^, median (IQR ^d^)	37 (30–44)	36 (29–44)	37 (30–44)
Unknown	27,839	10,136	17,703
**Dataset 2**			
		**Prepartum Vaccination Status**	
	**Overall** **(*n =* 1002)**	**NON VACC ^a^** **(*n =* 525)**	**VACC ^b^** **(*n =* 477)**
**SCC ^c^**, median (IQR ^d^)	49 (32–69)	47 (29–68)	50 (34–69)
**ECM 305 ^e^**, median (IQR ^d^)	8748 (7351–9911)	8696 (7190–9833)	8797 (7506–10,004)
**ECM FTD ^f^**, median (IQR ^d^)	29.9 (26.6–33.3)	29.8 (26.9–33.2)	30.0 (26.4–33.5)

^a^ no vaccination during the dry period; ^b^ vaccination between 2.5 and 8 weeks before expected calving date; ^c^ somatic cell count on the first day of milk recording (in thousand); ^d^ interquartile range; ^e^ Energy corrected milk yield in 305 days lactation in kg; ^f^ Energy corrected milk yield on the first day of milk recording in kg.

**Table 3 animals-15-00203-t003:** Association between prepartum vaccination status and mastitis prevalence in multivariable linear mixed-effects logistic regression.

Variables	OR ^a^	95% CI ^b^	*p*-Value ^c^
Prepartum vaccination status			
VACC ^d^/NON VACC ^e^	0.99	0.86, 1.14	0.869
Parity			
secondiparous/primiparous	0.73	0.66, 0.80	**<0.001**
multiparous/primiparous	1.30	1.20, 1.40	**<0.001**
multiparous/secondiparous	1.78	1.64, 1.94	**<0.001**
Access to pasture ^f^			
yes/no	1.85	1.61, 2.12	**<0.001**
Calving season			
summer/spring	1.32	1.21, 1.44	**<0.001**
autumn/spring	1.00	0.91, 1.10	>0.999
autumn/summer	0.76	0.70, 0.83	**<0.001**
winter/spring	0.96	0.87, 1.06	0.681
winter/summer	0.73	0.67, 0.79	**<0.001**
winter/autumn	0.96	0.87, 1.05	0.577

^a^ odds ratio for pairwise contrasts; ^b^ confidence interval; ^c^ significant effects are marked in bold-threshold: 0.0014; ^d^ vaccination between 2.5 and 8 weeks before expected calving date; ^e^ no vaccination during the dry period; ^f^ access to pasture during dry period; herd and calving year were applied as random effects.

**Table 4 animals-15-00203-t004:** Prepartum vaccination status and somatic cell count ^a^ in multivariable linear mixed-effects regression.

	Model Estimates	95% CI ^b^	*p*-Value ^c^
Prepartum vaccination status			
VACC ^d^–NON VACC ^e^	−0.06	−0.12, 0.00	0.036
Parity			
secondiparous–primiparous	−0.18	−0.22, −0.15	**<0.001**
multiparous–primiparous	0.23	0.20, 0.26	**<0.001**
multiparous–secondiparous	0.41	0.38, 0.44	**<0.001**
Flooring			
deeplitter–slatted floor	0.16	0.11, 0.21	**<0.001**
Herdsize ^f^			
small–medium	−0.20	−0.40, −0.00	0.050
large–medium	−0.31	−0.52, −0.10	**0.001**
large–small	−0.11	−0.36, 0.14	0.551
Calving season ^g^			
summer–spring	0.15	0.11, 0.19	**<0.001**
autumn–spring	−0.01	−0.05, 0.03	0.971
autumn–summer	−0.16	−0.19, −0.12	**<0.001**
winter–spring	−0.03	−0.07, 0.01	0.171
winter–summer	−0.18	−0.22, −0.14	**<0.001**
winter–autumn	−0.02	−0.06, 0.01	0.339

^a^ somatic cell count was logarithmically transformed; ^b^ confidence interval; ^c^ significant effects are marked in bold-threshold: 0.0014; ^d^ prepartum vaccination during the dry period; ^e^ no vaccination during the dry period; ^f^ annual total number of cows in herd, segmented into categories small [184, 987], medium [890, 1275], and large [1276, 2625]; ^g^ spring (March–May), summer (June–August), autumn (September–November), winter (December–February); herd and calving year were applied as random effects.

**Table 5 animals-15-00203-t005:** Prepartum vaccination status and energy corrected milk yield in multivariable linear mixed-effects logistic regression.

	ECM 305 ^a^			ECM FTD ^b^		
	**Model Estimates**	**95% CI ^c^**	** *p* ** **-Value ^d^**	**Model Estimates**	**95% CI ^c^**	** *p* ** **-Value ^d^**
**Prepartum vaccination status**						
VACC ^e^–NON VACC ^f^	5.0	−224, 234	0.966	−0.42	−0.99, 0.15	0.148
Replacement rate	−65	−88, −42	**<0.001**	−0.15	−0.21, −0.09	**<0.001**
Rest period ^g^	14	8.3, 21	**<0.001**			

^a^ energy corrected milk yield in 305 days of lactation; ^b^ energy corrected milk yield on first day of milk recordings; ^c^ confidence interval; ^d^ significant effects are marked in bold-threshold: 0.0158; ^e^ prepartum vaccination during the dry period; ^f^ no vaccination during the dry period; ^g^ rest period was only applied for response variable ECM 305 due to temporal overlap with ECM FTD; herd was applied as random effect.

## Data Availability

The original contributions presented in this study are included in the article/Appendix A. Further inquiries can be directed to the corresponding author.

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
