# Peer review of "Prepartum Vaccination Against Neonatal Calf Diarrhea and Its Effect on Mammary Health and Milk Yield of Dairy Cows: A Retrospective Study Addressing Non-Specific Effects of Vaccination"

_animals, 2025, doi:10.3390/ani15020203_

Round 1
Reviewer 1 Report
Comments and Suggestions for Authors
Review of Manuscript ID - animals-3390249
Prepartum vaccination has no influence on mammary health and milk yield of dairy cows: a retrospective study addressing non-specific effects of vaccination
General comments
This is generally a well written paper addressing a potentially important issue. Although the direct adverse effects of vaccination have been examined as part of vaccine registration studies the non-specific impacts have been poorly researched. The authors raised the issue of epigenetic effects on neonatal immune function but did not investigate this in their study – an explanation is required. Also, it needs to be acknowledged that this study examined only the impact of vaccination of late pregnant, non-lactating cows. It is not uncommon for lactating cows to be vaccinated. The process of vaccination is stressful for cattle and there is potentially an increased risk of udder infections due to stress induced immune suppression and disturbances in teat sphincter closure when cows are vaccinated immediately after milking. A brief review of the short term changes in cow physiology and immune function following vaccination should be included in the introduction and briefly discussed int discussion section.
Specific comments
Title should specifically state that vaccination was against neonatal calf diarrhea pathogens
Ln 42 – consider deleting ‘emphasizing the importance of herd management’ or rewording as does not make sense
Ln 49 – replace ‘mother’ with ‘cow’
Ln 60 – evidence to support statement required
Ln 95 – replace ‘leveled’ with balanced
Ln 124 – ‘wrangling’ – not internationally recognised terminology – replace please
Ln 158 – ‘was subset.’ – reword does not make sense
Figure 1 – font too small
Section 2.4, 2.5 – consider replacing ‘elaboration’ with ‘description’
Ln 397 – ‘although the’ should be ‘although they’
Author Response
Dear Reviewer,
we sincerely thank you for your thoughtful comments and constructive suggestions, which have significantly helped us refine our manuscript. We have carefully addressed each of your points and made the corresponding adaptation, as detailed below. We hope these changes meet your expectations and enhance the scientific quality and clarity of the manuscript.
Kind regards in the name of all authors,
Caroline Kuhn
Comment 1: This is generally a well written paper addressing a potentially important issue. Although the direct adverse effects of vaccination have been examined as part of vaccine registration studies the non-specific impacts have been poorly researched. The authors raised the issue of epigenetic effects on neonatal immune function but did not investigate this in their study – an explanation is required. Also, it needs to be acknowledged that this study examined only the impact of vaccination of late pregnant, non-lactating cows. It is not uncommon for lactating cows to be vaccinated. The process of vaccination is stressful for cattle and there is potentially an increased risk of udder infections due to stress induced immune suppression and disturbances in teat sphincter closure when cows are vaccinated immediately after milking. A brief review of the short term changes in cow physiology and immune function following vaccination should be included in the introduction and briefly discussed int discussion section.
Response 1: Indeed, we mention epigenetic alterations in immune function in utero (Ln 67-69), referencing a study conducted in mice. We included this study to highlight, that innate immune changes after vaccination during pregnancy have been previously demonstrated. It was one of the few studies addressing these effects during pregnancy. However, we acknowledge that the wording in the manuscript may have been misleading regarding the effects on the fetus. To clarify, we have revised the text to focus more explicitly on the changes occurring within the mother during pregnancy: “The innate response to vaccination during pregnancy has been demonstrated in mice, where the development of the fetal immune system was modified in utero by epigenetic changes. This suggests that vaccination-induced epigenetic modification also alters the immune reactivity of the vaccinated mother” (Ln 67-71). Investigating the effects of vaccination on the immune function of the calf, beyond the known beneficial effects via colostrum quality, would indeed have been highly interesting. However, this was beyond the scope of the current study.
We also acknowledge that lactating cows are vaccinated, and you raise an important point regarding the potential stress-related risks, such as increased susceptibility to udder infections and disturbances in teat sphincter closure when cows are vaccinated immediately after milking. However, the focus of this study was on the vaccination of late-pregnant, non-lactating cows. The prepartum vaccine against neonatal calf diarrhea, as recommended by the manufacturers, is to be administered between three and 12 weeks before the expected calving date. In this study, all participating farms vaccinated their cows no earlier than eight weeks before the expected calving date. Consequently, all cows included in this study were already dry at the time of vaccination, and it was not possible to investigate non-specific effects on prepartum lactation and prepartum udder health.
Short term changes in the context of adverse effects after vaccination were added to the introduction of the manuscript: “Prepartum vaccination of cows is regarded as safe and usually no adverse reactions are reported. Adverse reactions are known as any harmful or unintended outcomes that are directly caused by the vaccine or its components, including hypersensitivity, fever, local swelling, and short-term reactions such as increased levels of acute phase proteins. These reactions can differ between live and non-live vaccines, as well as be-tween adjuvants administered.” (Ln 51-56). We assumed that short-term effects did not have an impact on the response variable in the postpartum period, as they had already faded by that time. Therefore, we didn’t include them in the discussion.
Comment 2: Title should specifically state that vaccination was against neonatal calf diarrhea pathogens
Response 2: We changed the title into: Prepartum vaccination against neonatal calf diarrhea and its effect on mammary health and milk yield of dairy cows: a retrospective study addressing non-specific effects of vaccination.
pComment 3: Ln 42 – consider deleting ‘emphasizing the importance of herd management’ or rewording as does not make sense
Response 3: Thank you for pointing this out. We have deleted the clause.
Comment 4: Ln 49 – replace ‘mother’ with ‘cow’
Response 4: We agree and replaced "mother" with "cow" in line 49. Also, we have ensured that this change is consistently applied throughout the manuscript where applicable (Ln 19).
Comment 5: Ln 60 – evidence to support statement required
Response 5: addressed in Response 1.
Comment 6: Ln 95 – replace ‘leveled’ with balanced
Response 6: the wording has been adjusted.
Comment 7: Ln 124 – ‘wrangling’ – not internationally recognised terminology – replace please
Response 7: Thanks for pointing this out! This term originally comes from R programming and is indeed unscientific. We have updated it to “pre-processing” in all three locations to ensure it aligns with more universally accepted scientific terminology (Ln 130, 168, 476).
Comment 8: Ln 158 – ‘was subset.’ – reword does not make sense
Response 8: right, we replaced it with ‘isolated'.
Comment 9: Figure 1 – font too small
Response 9: We have increased the font size to address the issue. As the original tool could not accommodate the necessary adjustments, we recreated the flowchart using R. While the appearance has slightly changed, we believe this modification has further improved the clarity and readability of the figure.
Comment 10: Section 2.4, 2.5 – consider replacing ‘elaboration’ with ‘description’
Response 10: wording has been adjusted.
Comment 11: Ln 397 – ‘although the’ should be ‘although they’
Response 11: To improve clarity, we revised the sentence to: “These findings mirror in part those of Scott et al. [16], who found no effect on milk yield after vaccination with a core antigen vaccine against gram-negative bacteria, although it contained a dedicated immune-enhancing/-modulating adjuvant (Immune Plus®). Our findings argue against a non-specific effect of prepartum cow vaccination with the NCD vaccines applicated” (Ln 401-405).
Reviewer 2 Report
Comments and Suggestions for Authors
The current investigation provides relevant information regarding non-specific effects of booster vaccinations in pregnant cows. However, the simple summary, the abstract, and introduction, discussion, and conclusion sections should be reorganized in a logical order. The concept of non-specific effects of vaccination is only introduced in line 68 and the connection with prepartum vaccination of cows against neonatal calf diarrhea is only made in lines 82 and 83, making it hard for the reader to understand the context and goals of your research.
Besides, how do references 6, 9, 10, 11, and 12 relate to cattle? I know that the placenta of the cow is sindesmocorial, insted in the rodents and humans the placenta is hemocorial. I desagree with the authors, it is not apropriated to compare results of papers about humans and rodents´ placentas with the cow specie.
Line 15. Cut repetition “...Unterschleissheim, Germany, Germany...”
Lines 18 and 19 are out of sync with lines 30 and 31. Suggestion for lines 18 and 19:
“Prepartum vaccination of cows are performed to boost colostral antibodies available to calves in the first weeks of life.”
Line 24. Eastern Germany refers to the former East Germany. Write in lowercase or name the region.
Line 42. What is the meaning of “non-specific effects”? The authors wanted to demonstrate that prepartum vaccination does not alter the incidence of mastitis, somatic cell counts and the volume of milk produced. All that can be said is that these parameters did not show any statistical difference.
Lines 43 and 44. In the same sense, what are “non-specific effects”? The correct thing to do would be to investigate the adverse effects of prepartum vaccination.
Lines 48, 49, and 50. Rearrange and merge the two sentences to make the question clearer. The justification for vaccinating the cow before calving is so that it will produce more colostrum, and it is this colostrum enriched by the cow's vaccine response that will provide additional protection to the calf.
Lines 53 and 54. What is the reason for comparing animals with such different placentas?
Line 99. “... in the former DDR...”
Lines 118, 119, and 120. The expression “attendant veterinarians” does not make much sense. The authors probably refer to veterinarians who regularly attend the farms recruited for the investigation. Do these veterinarians work for the RA or are they local clinicians?
Line 231. Dry Cow Therapy is more common.
Line 259. Proc GLM. Does “Generalized” need to be in parentheses?
Line 267. An article is missing: “... the p-value”.
Lines 271, 272 and 273. An element is missing from the sentence.
Line 315. “Results” section, the title of item 3.2 anticipates the conclusion.
Captions for Tables 3 and 4. A “C” is missing in “onfidence interval”.
Line 330. The title of the item anticipates the conclusion.
Line 372. The term “postpartal” seems incorrect to me. The adjective is “postpartum”.
Lines 372 to 388. This paragraph seems to stray from the objective of the study. What is the problem? Is the objective of vaccination to benefit the calf by increasing the supply of antibodies in the colostrum or to modulate the cow’s immune system to prevent diseases in the pre-partum period?
Line 397. “...used vaccine...” It is not an adjective, you are referring to the vaccines USED in the study.
Line 432. “...a robust collinear model...”
Lines 490 and 491. Are SCC and milk yield records automated?
CONCLUSION: the paragraph should be moved to DISCUSSION, it should be the last paragraph in DISCUSSION
Suggestion for conclusion: The prepartum vaccination of cows with neonatal calf diarrhea using three different commercial products didn´t affect milk production, somatic cell count, and the incidence of mastitis.

There are some corrections in English language to do.
But the most important is to reorganize the paragraphs in a sequencial form.
Author Response
Dear Reviewer,
we sincerely thank you for your thoughtful comments and constructive suggestions, which have significantly helped us refine our manuscript. We have carefully addressed each of your points and made the corresponding adaptation, as detailed below. We hope these changes meet your expectations and enhance the scientific quality and clarity of the manuscript.
Kind regards in the name of all authors,
Caroline Kuhn
Comment 1: The current investigation provides relevant information regarding non-specific effects of booster vaccinations in pregnant cows. However, the simple summary, the abstract, and introduction, discussion, and conclusion sections should be reorganized in a logical order. The concept of non-specific effects of vaccination is only introduced in line 68 and the connection with prepartum vaccination of cows against neonatal calf diarrhea is only made in lines 82 and 83, making it hard for the reader to understand the context and goals of your research.
Besides, how do references 6, 9, 10, 11, and 12 relate to cattle? I know that the placenta of the cow is sindesmocorial, insted in the rodents and humans the placenta is hemocorial. I desagree with the authors, it is not apropriated to compare results of papers about humans and rodents´ placentas with the cow specie.
Response 1: Thank you for your valuable feedback. We agree that the structure of the introduction and the integration of the concept of non-specific effects (NSE) needed improvement to enhance clarity and logical flow. To address this, we have restructured the introduction to provide a clearer focus on NSE and its mechanisms, ensuring that the concept is introduced earlier and in a more structured manner. This includes a better explanation of trained immunity/innate memory as components of NSE. Also, we have added neonatal calf diarrhea (NCD) to the first sentence to establish its relevance to the study's context from the beginning.
Your comment regarding the comparability of studies conducted in species other than cows is a very interesting point. You raise a valid point regarding the challenges in comparing studies across species. Unfortunately, there is a limited number of studies investigating non-specific effects of vaccines specifically in cows (compare Ln 63-65, 78-80). To address this gap, we included studies involving other species to provide context and highlight broader insights into non-specific effects. This also aligns well with the aims and scopes of the journal Animals (“to provide an understanding of animals within a larger context”), which we understood that the comparison with other species is appreciated. However, we recognize the importance of emphasizing species-specific differences and have revised the text for clarity. Therefore, we have added the statement “Progress has been made in understanding NSEs, particularly in rodents and humans” (Ln 64-65), to acknowledge that much of the existing knowledge comes from studies in these species. Furthermore, we have deleted the sentence comparing prepartum vaccination in women against SARS-CoV-2 (formerly Ln 53–54), including the associated citation of Leik et al. (2022), in alignment with another reviewer’s comment. Regarding the study on in utero fetal immune function by Amir and Zeng (2021), please refer to our clarifications and adaptations in our response to Comment 8. We appreciate your input and have worked to improve the text, ensuring a more appropriate and species-specific discussion where possible. Let us know if further adjustments are needed.
Comment 2: Line 15. Cut repetition “...Unterschleissheim, Germany, Germany...”
Response 2: The repetition was deleted.
Comment 3: Lines 18 and 19 are out of sync with lines 30 and 31. Suggestion for lines 18 and 19: “Prepartum vaccination of cows are performed to boost colostral antibodies available to calves in the first weeks of life.”
Response 3: We revised the text accordingly.
Comment 4: Line 24. Eastern Germany refers to the former East Germany. Write in lowercase or name the region.
Response 4: Thank you for pointing out this issue. The correct phrasing could indeed be either "former East Germany" as the historical region or "(north-)east Germany" for a geographic distinction. The confusion arose due to the different handling of these terms in German. Since the historical aspect is not particularly relevant in the abstract and summary, and a broader geographic reference suffices, we have simplified it to "Germany" in this context. In Line 106, where further distinction is necessary, we have clarified it as the "former East Germany" (see Response 9).
Comment 5: Line 42. What is the meaning of “non-specific effects”? The authors wanted to demonstrate that prepartum vaccination does not alter the incidence of mastitis, somatic cell counts and the volume of milk produced. All that can be said is that these parameters did not show any statistical difference.
Response 5: You are correct that the term "non-specific" might be unnecessary in this context. We have revised the sentence in the abstract to improve clarity and focus. It now reads: “The study concludes that prepartum vaccination against calf diarrhea has no significant effects on mammary health and milk yield” (Ln 40-41).
Comment 6: Lines 43 and 44. In the same sense, what are “non-specific effects”? The correct thing to do would be to investigate the adverse effects of prepartum vaccination.
Response 6: The aim of the study was to investigate potential non-specific effects (NSE), a term which differs from adverse effects. While adverse effects are harmful events that are directly caused by the vaccine or its components, such as hypersensitivity, fever, local swelling, and acute phase proteins, NSEs can be positive or negative. NSEs are mainly in opposition to antigen-specific effects and are mainly related to innate immune responses. Given the word limit of the abstract (max. 200 words), we are constrained in providing a more detailed explanation in the abstract, therefore we have elaborated on the terms and definitions in the main text of the introduction: “Prepartum vaccination of cows is regarded as safe and usually no adverse reactions are reported. Adverse reactions are known as any harmful or unintended outcomes that are directly caused by the vaccine or its components, including hypersensitivity, fever, local swelling, and short-term reactions such as increased levels of acute phase proteins. These reactions can differ between live and non-live vaccines, as well as be-tween adjuvants administered. In contrast to the adverse effects of vaccination, which are harmful, there are vaccination-induced effects beyond antigen-specific in-duction of adaptive immune responses, known as non-specific effects (NSEs). These can be beneficial or harmful for the individual. [...]” (Ln 51 ff.). We hope that this enhances the clarity of the terms in the context of this study.
Comment 7: Lines 48, 49, and 50. Rearrange and merge the two sentences to make the question clearer. The justification for vaccinating the cow before calving is so that it will produce more colostrum, and it is this colostrum enriched by the cow's vaccine response that will provide additional protection to the calf.
Response 7: We have revised and merged the sentences to improve clarity. The updated text now reads: “Prepartum vaccinations of cows are primarily performed to protect calves from infectious diseases in their first weeks of life. For this purpose, the cow is vaccinated during the last weeks of pregnancy to induce an adaptive immune response with high levels of pathogen-specific antibodies and obtain enriched colostrum for additional protection of the newborn calf” (Ln 47-51).
Comment 8: Lines 53 and 54. What is the reason for comparing animals with such different placentas?
Response 8: Animals with different types of placentas can transfer information to the fetus via stimulus/stressor-induced mediators introducing epigenetic changes in the fetus. For instance, this has been shown for cattle (e.g., heat stress, transport). We included this study to highlight, that innate immune changes after vaccination during pregnancy have been previously demonstrated as one of these stressors. It was one of the few studies addressing these effects during pregnancy. However, we acknowledge that the wording in the manuscript may have been misleading regarding the effects on the fetus via the placenta. Of course, the hemochorial placenta of rodents is not comparable to the epitheliochorial placenta of ruminants. To clarify, we have revised the text to focus more explicitly on the changes occurring within the mother during pregnancy: “The innate response to vaccination during pregnancy has been demonstrated in mice, where the development of the fetal immune system was modified in utero by epigenetic changes. This suggests that vaccination-induced epigenetic modification also alters the immune reactivity of the vaccinated mother” (Ln 67-71).
Comment 9: Line 99. “... in the former DDR...”
Response 9: replaced by ‘former East Germany’ (see Response 4).
Comment 10: Lines 118, 119, and 120. The expression “attendant veterinarians” does not make much sense. The authors probably refer to veterinarians who regularly attend the farms recruited for the investigation. Do these veterinarians work for the RA or are they local clinicians?
Response 10: We agree that the term “attendant veterinarians” is not very common in English. We have clarified this by changing the term to “local regularly visiting veterinarians” (Ln 125, 126).
Comment 11: Line 231. Dry Cow Therapy is more common.
Response 11: We replaced the term.
Comment 12: Line 259. Proc GLM. Does “Generalized” need to be in parentheses?
Response 12: Thank you for pointing this out. We have removed the parentheses since both models are generalized. For clarification: our original intention was to differentiate between logistic and linear mixed-effects models, though the use of the parentheses was not ideal. However, as both types of models are generalized mixed-effects models with different distribution families, we now specify them as follows:
- When the response variable is binary and follows a Binomial distribution, we refer to it as a logistic mixed-effects model.
- When the response variable is numeric and follows a Gaussian distribution, we refer to it as a linear mixed-effects model.
We hope this explanation provides clarity. Please let us know if further adjustments are needed.
Comment 13: Line 267. An article is missing: “... the p-value”
Response 13: the article was added to the manuscript.
Comment 14: Lines 271, 272 and 273. An element is missing from the sentence.
Response 14: Indeed, the sentence was not complete. We adapted as follows: “When preparing the multivariable analysis for the response variables ECM FTD and ECM 305, it soon became clear, that variables other than the vaccination-related parameters overlapped the results” (Ln 277-279).
Comment 15: Line 315. “Results” section, the title of item 3.2 anticipates the conclusion.
Response 15: We intentionally framed the subtitles in sections 3.2, 3.3, and 3.4 to align with the conclusions; however, we agree that they were overly conclusive. To address this, we have rephrased the subtitles with a stronger focus on the results as follows:
- 3.2. "Prepartum vaccination was not significantly associated with mammary health parameters" (Ln 321)
- 3.3. “Prepartum vaccination was not significantly associated with milk yield in healthy primiparous cows" (Ln 336)
- 3.4. “Herd management related factors were most relevant for mammary health” (Ln 361)
Comment 16: Captions for Tables 3 and 4. A “C” is missing in “onfidence interval”.
Response 16: We corrected this typo.
Comment 17: Line330. The title of the item anticipates the conclusion.
Response 17: addressed in Response 15.
Comment 18: Line 372. The term “postpartal” seems incorrect to me. The adjective is “postpartum”.
Response 18: we corrected it here and in Ln 81.
Comment 19: Lines 372 to 388. This paragraph seems to stray from the objective of the study. What is the problem? Is the objective of vaccination to benefit the calf by increasing the supply of antibodies in the colostrum or to modulate the cow’s immune system to prevent diseases in the pre-partum period?
Response 19: Thank you for this insightful comment, which we have thoroughly discussed in the project team previously. The primary objective of vaccination against neonatal calf diarrhea (NCD) is indeed to improve colostrum quality to protect the calf. However, as outlined in the aims of the study (Ln 86–88), our focus is on the potential non-specific effects of prepartum vaccination on the cow itself. We recognize that this distinction was not entirely clear. To address this, we have clarified our hypothesis in the introduction as follows:
“We hypothesized that the prepartum vaccination against NCD induces beneficial non-specific effects, which allow the dysregulated immune function of the periparturient cow to better cope with mammary infections. These effects would be measurable through improved mammary health outcomes and production metrics during the very sensitive peripartum period” [Ln 88-92].
We hope this addition resolves any ambiguity and better aligns the study's objective with its focus on the cow.
Comment 20: Line 397. “...used vaccine...” It is not an adjective, you are referring to the vaccines USED in the study.
Response 20: To improve overall clarity of the sentence, we have revised the sentence as follows: “These findings mirror in part those of Scott et al., who found no effect on milk yield after vaccination with a core antigen vaccine against gram-negative bacteria, although it contained a dedicated immune-enhancing/-modulating adjuvant (Immune Plus®). Our findings argue against a non-specific effect of prepartum cow vaccination with the NCD vaccines applicated” (Ln 401-405). With this revision, the term "used" is no longer necessary. We hope this addresses your concern effectively.
Comment 21: Line 432. “...a robust collinear model...”
Response 21: The phrase was intended to indicate that the Random Forest algorithm is robust in cases where variables in the dataset are multicollinear, rather than suggesting that Random Forest itself is a collinear model. To improve clarity, we have revised the text as follows: “The Random Forest machine learning-algorithms, a model robust to multicollinearity of variables, facilitated the ranking of importance of influencing factors” (Ln 437-439).
Comment 22: Lines 490 and 491. Are SCC and milk yield records automated?
Response 22: Yes, the SCC and milk yield records were automated on the participating farms. We have clarified this in the manuscript.
Comment 23: CONCLUSION: the paragraph should be moved to DISCUSSION, it should be the last paragraph in DISCUSSION. Suggestion for conclusion: The prepartum vaccination of cows with neonatal calf diarrhea using three different commercial products didn´t affect milk production, somatic cell count, and the incidence of mastitis
Response 23: We transferred parts of the conclusion at the end of the discussion, incorporated the suggested sentence at the beginning and streamlined the conclusion (Ln 514ff.).
Comment 24: There are some corrections in English language to do. But the most important is to reorganize the paragraphs in a sequencial form
Response 24: We hope that the overall language and reorganization of conclusion statements is now more clear.
Reviewer 3 Report
Comments and Suggestions for Authors
This is a retrospective study of the impact of prepartum vaccination against neonatal calf diarrhea on milk yield and mammary health mammary in cows (73,378 cows from 20 farms).
The title appears overly direct, please modify it a bit, leaving room for further investigations
Please delete this phrase: Vaccinations of pregnant women against SARS-CoV-2 also revealed no harmful effects on pregnancy.
Please insert your hypothesis after the aim
Please replace monitoring during birth with during calving
Please expand your limitations
Author Response
Dear Reviewer,
we sincerely thank you for your thoughtful comments and constructive suggestions, which have significantly helped us refine our manuscript. We have carefully addressed each of your points and made the corresponding adaptation, as detailed below. We hope these changes meet your expectations and enhance the scientific quality and clarity of the manuscript.
Kind regards in the name of all authors,
Caroline Kuhn
Comment 1: The title appears overly direct, please modify it a bit, leaving room for further investigations
Response 1: We changed the title into: Prepartum vaccination against neonatal calf diarrhea and its effect on mammary health and milk yield of dairy cows: a retrospective study addressing non-specific effects of vaccination.
Comment 2: Please delete this phrase: Vaccinations of pregnant women against SARS-CoV-2 also revealed no harmful effects on pregnancy
Response 2: Thank you for your feedback. We acknowledge that the phrase was not closely related to the context. As a result, we have deleted it to improve the relevance and focus of the text.
Comment 3: Please insert your hypothesis after the aim
Response 3: We have added our hypothesis, which aligns well with the first paragraph of the discussion section:
“We hypothesized that the prepartum vaccination against NCD induces beneficial non-specific effects, which allow the dysregulated immune function of the periparturient cow to better cope with mammary infections. These effects would be measurable through improved mammary health outcomes and production metrics during the very sensitive peripartum period” (Ln 89-93).
Comment 4: Please replace monitoring during birth with during calving
Response 4: We replaced the wording.
Comment 5: Please expand your limitations
Response 5: In the discussion section of the manuscript, we addressed several limitations of our study and their potential impact on the findings. One key limitation was the variability in health documentation across farms, which may have led to misclassification or inconsistencies of diagnoses (see Ln 460ff.). To address this, we implemented extensive pre-processing of the datasets, including exploratory analyses of herd-specific documentation patterns, comprehensive models and integration of robust variables, such as milk yield parameters from automated milk testing to ensure comparability (see Ln 496-499). The retrospective nature of the data posed additional challenges, as documentation practices evolved over time, and we recognized the potential bias of farm management practices influencing vaccination decisions (see Ln 469-470). We mitigated this by analyzing herd management-related variables and incorporating robust statistical methods. Although we focused on farms with a shared historical background to improve comparability, heterogeneity in management and diagnostic practices could not be completely eliminated. Furthermore, we highlighted the nature of large datasets, which, while enhancing statistical power, increase the risk of type I errors. Here, we applied a stricter p-value threshold (see Ln 451-457). Additionally, the variability in the timing of milk testing across farms and the retrospective nature of the data required careful interpretation, but we concluded that this variability did not significantly affect our results after analyzing milk performance trends (see Ln 507-510).
With regards to your comment, we admit, that the limitations of the study could be discussed in more detail. Therefore, we have expanded them as follows:
“These visits were usually accompanied by the local veterinarians, who could provide an assessment of herd management over the past years and therefore beyond the time of the onsite-survey”. (Ln 478-480)
“[...] but not entirely eliminating heterogeneity in management practices”. (Ln 483)
“However, the retrospective nature of the data and the reliance on farm-level docu-mentation practices remain limitations that could affect accuracy”. (Ln 489-490)
“Nevertheless, the potential for unmeasured confounders cannot be completely ruled out”. (Ln 499-500)
“we concluded that the differing timing of data collection did not significantly impact the results. Despite these measures, the retrospective nature and inherent variability of the dataset introduce some limitations to the study’s explanatory power. Moreover, the scope of the study was limited to three specific vaccine products and to the mammary gland, which may not capture all effects of other prepartum vaccinations or impacts on other organ systems” (Ln 509-519).
We hope this response clarifies how we addressed the limitations and appreciate the opportunity to further refine our discussion. We hope that these adaptations meet the expectations.
Round 2
Reviewer 2 Report
Comments and Suggestions for Authors
In scientific language, we don´t use personal pronoums. Instead, use passive voice in lines:
89
157
194
198
258
267
269
395
455
476
478
484
486
500
506
509
These suggestions are written inside the ballons.

Author Response
Dear Reviewer,
thank you very much for your prompt second review and for pointing out the personal pronouns. We have revised all of the specified lines and have rephrased them in the passive voice to align with scientific writing conventions. Please find the updated text in the revised manuscript.
Sincerely,
Caroline Kuhn